# Lost to follow-up and associated factors among patients with drug resistant tuberculosis in Ethiopia: A systematic review and meta-analysis

Assefa Andargie[ID]1*, Asressie Molla1, Fentaw Tadese1, Segenet Zewdie2

1 School of Public Health, College of Medicine and Health Sciences, Wollo University, Dessie, Ethiopia,
2 Department of Pharmacy, College of Medicine and Health Sciences, Wollo University, Dessie, Ethiopia

* assefaand@gmail.com

**Data Availability Statement:** All relevant data are within the paper and its Supporting information files.

## Abstract

### Background

One third of global antmicrobial resistance deaths are attributed to drug resistant tuberculosis. Lost to follow-up is one of the causes of the development of acquired drug resistant tuberculosis. There is a gap in nationally representative reliable information on lost to follow-up among patients with drug-resistant tuberculosis in Ethiopia.

### Objective

To estimate the pooled prevalence and associated factors of lost to follow-up among patients with drug resistant tuberculosis in Ethiopia.

### Methods

Observational studies searched from PubMed, HINARI and CINAHL were screened for eligibility. After assessing the quality of studies, data were extracted using a checklist. Heterogeneity was assessed using forest plot, Q and $I^2$. The random effects meta-analysis model was employed to pull the prevalence of lost to follow-up. Sub-group analysis and meta regression were performed to identify the sources of heterogeneity. Publication bias was assessed using funnel plots with Egger's and Begg's tests. Sensitivity analysis was performed to assess the influence of individual studies on the overall estimate. The odds ratios were used to measure associations.

### Results

The review was performed among 11 studies of which 9 were cohort studies. The sample sizes ranged from 90 to 612 and comprised a total of 3,510 participants. The pooled prevalence of lost to follow-up was 8.66% (95% CI, 5.01–13.14) with a high heterogeneity ($I^2$ = 93.49%, p<0.001). Pulmonary multi-drug resistant tuberculosis patients were 50% less likely to loss from follow-up compared to extra pulmonary tuberculosis patients.

**Funding:** The authors received no specific funding for this work.

**Competing interests:** The authors have declared that no competing interests exist.

## Conclusion

There was a high prevalence of lost to follow-up among multi-drug resistant tuberculosis patients in Ethiopia. Anatomical site of tuberculosis was a significant factor affecting lost to follow-up. Strengthening the health care system and patient education should be given a due emphasis.

## Registration number

CRD42020153326; https://www.crd.york.ac.uk/prospero/display_record.php?RecordID=153326.

## Introduction

Drug resistant tuberculosis (DR-TB) is an emerging global public health threat. In 2018, 484,000 people developed TB that was resistant to rifampicin (RR-TB), and of these, 78% had multidrug-resistant TB (MDR-TB). About 187,000 cases of MDR/RR-TB were detected and notified in this year. Among cases of MDR-TB in the same year, 6.2% were estimated to have extensively drug-resistant TB (XDR-TB) [1]. In the year 2017, one-third of global antimicrobial resistance deaths are attributed to MDR-TB [2].

Ethiopia is among the 30 high TB/DR-TB burden countries around the globe. According to a study based on reference laboratory data, the overall prevalence of DR-TB in Ethiopia was 11.6% [3]. Programmatic management of DR-TB was first initiated in 2009 in Ethiopia [4]. Currently, there are three nationally recommended DR-TB treatment regimens in Ethiopia; Standardized Longer Treatment Regimen (sLTR), Standardized Shorter Treatment Regimen (sSTR) and Individualized longer Treatment Regimen (ITR) [5].

Based on the treatment outcome, a TB patient may be categorized as cured, treatment completed, treatment failure, loss to follow up or died [6]. Lost to follow up (LTFU), formerly known as treatment default, refers to patients who received treatment for at least 4 weeks and discontinued treatment for more than eight consecutive weeks [5,7]. The prevalence of patients with LTFU varies in different parts of the world. According to a systematic review in Sub Saharan Africa a range of 1% to 22.3% of DR-TB patients were LTFU [8].

LTFU is one of the reasons for the development of acquired DR-TB. Patients who return to retreatment after LTFU were at a higher risk of developing DR-TB and are at a higher risk for poor prognosis and death [9–11]. Reports on economic burden of non-adherence to TB medicines indicated that an estimated 52 MDR-TB patients who were lost to follow-up are likely to have resulted in 5 patients developing XDR-TB, 3 new persons being infected with MDR-TB and 1 new person with XDR-TB, and 3 deaths. Moreover, LTFU is likely to have resulted in nearly USD 380,000 in additional costs (USD 325,000 in health service costs and USD 55,000 in household and society costs). This is equivalent to a cost of more than USD 7,000 per patient [12].

There are many primary studies conducted on the treatment outcome and associated factors of DR-TB. These primary studies provided variations in the estimate of the prevalence and associated factors of LTFU among DR-TB patients. Some systematic reviews and meta-analyses were published on poor treatment outcomes [13,14]. However, the outcome measurement in the review published by Eshetie S. et al included unevaluated patients in the computation of LTFU. Our review excluded those patients whose treatment outcome was 'not

evaluated' (including transfer outs and patients still on treatment). To determine the final treatment outcome of a patient, that patient should go through the recommended regimen duration. As such, the prevalence of LTFU may be under estimated in the mentioned review article due to large size of the denominator. The other review by Tola H. et al was concerned about treatment non adherence (including intermittent treatment and LTFU together) among any TB patients, not only among DR-TB patients [14].

Other systematic reviews generally considered factors associated with good or poor treatment outcomes. Here the poor outcome included death, treatment failure, treatment default and/or LTFU together. There is no pooled evidence on the factors associated, specifically, with treatment LTFU which is an important outcome related to the development of acquired drug resistance and of course playing an important role in the transmission of DR-TB within the community. Therefore, this review was aimed at estimating the prevalence of LTFU and associated factors among patients with MDR-TB in Ethiopia.

## Methods

### Study design

A systematic review and Meta-analysis of published and unpublished studies was used to determine pooled prevalence of LTFU and its predictors among MDR-TB patients in Ethiopia. The review was reported in accordance with the Preferred Reporting Items for Systematic Reviews and Meta-Analyses (PRISMA) Statement [15].

### Protocol and registration

Initially, databases were searched to check for the same systematic review in order to avoid duplicates. PubMed, Cochrane/Wiley Library and the international prospective register of systematic review and meta-analysis (PROSPERO) were explored to confirm whether previous systematic review and/or meta-analyses existed with the same topic. This systematic review and meta-analysis were registered at PROSPERO with registration number of CRD42020153326.

### Inclusion criteria

The inclusion criteria were delimited using the PECOS components.

- **Patient**: patients with drug resistant tuberculosis by which their treatment outcome was known and reported based on the national DR TB treatment guideline [5].

- **Exposure**: Being on MDR-TB treatment

- **Comparison**: No

- **Outcome**: lost to follow-up from MDR TB treatment defined as treatment interruption for two or more consecutive months (8 weeks) for any reason without medical approval [7].

- **Study design**: Observational studies including cross-sectional and cohort studies.

    **Time and language**: Articles published from 2009-May 2020 were included. The year 2009 is the year when Ethiopia started DR-TB treatment within a separate treatment center.
    **Geographical location**: studies conducted exclusively in Ethiopia

### Exclusion criteria

- Studies which were conducted exclusively among special populations like children, HIV/AIDS patients or only among patients with other comorbid conditions.

- Studies which provide only interim outcomes (defined as ≤9 months for shorter treatment regimen and ≤18 months for longer treatment regimen) to avoid bias towards a lower LTFU.

- Studies conducted among fewer than 50 participants

- Studies in which all patients had XDR TB or mono resistant TB

## Information sources

To access published primary studies PubMed, Health Inter Network Access to Research Initiative (HINARI) and CINAHL database sources were extensively searched. To supplement the electronic data base searches, the online archives of the International Journal of Tuberculosis and Lung Disease was searched for applicable studies. Grey literatures were retrieved using Google and google scholar electronic search engines. Research repository sites in Ethiopia like Addis Ababa University and National Academic Digital Library of Ethiopia (NADLE) were also searched. Following a snowballing system, the reference lists of the retrieved studies were probed to collect articles that are not accessible through databases as well as electronic search engines. The correspondence authors were contacted via e-mail for articles with incomplete information and responses were waited for a month. The last literature search was performed on May 20, 2020. We subscribed updates of new articles from PubMed and International Journal of Tuberculosis and Lung Disease. These updates were awaited until June 17, 2020.

## Search strategy

We searched for articles that included any combination of the following search terms in their singular or plural form in their title, abstract, keywords, and text: "treatment outcome", "poor treatment outcome", "treatment default", "lost to follow-up", "unfavorable treatment outcome", "multi drug resistant tuberculosis", "MDR-TB", "drug resistant tuberculosis" and "Ethiopia". The search was restricted to human species and published since 2009 limiters. The Boolean operators like "OR" and "AND" were used to combine terms and form the search syntax (S1 File).

## Study selection

The article searches and screening activity was done by two reviewers (AA and SZ). Articles were exported and managed using EndNote V.7, duplicates were identified and removed. The remaining articles were evaluated for eligibility by topic, abstract and full text levels. Unrelated topics and studies conducted out of Ethiopia were rejected. The abstracts and full texts of the remaining studies were reviewed. Those articles with incomplete information, by which their authors were unable to provide the missed information through e-mail contact, were excluded from the review.

## Data extraction process

We developed a data extraction sheet using Microsoft excel worksheet, pre-tested on five randomly-selected included studies, and modified the checklist accordingly. One review author (AA) extracted the data from included studies and the second author (SZ) checked the extracted data. Disagreements were resolved by discussion between the two review authors and there was no unresolved disagreement. We contacted the corresponding authors of selected studies through e-mail to get additional data on the factors associated with the LTFU.

After waiting for one month, only three authors responded and provided numerical data that has been used to compute effect sizes.

## Data items

Information was extracted from each included study on:

1. Study characteristics such as the name of correspondence author, publication year, study year, study setting/region/institution, sample size, study design, primary outcome and quality score

2. Participant characteristics such as sex, HIV status, comorbidity, BMI, previous treatment history, treatment category and anatomical site of TB

3. Outcomes such as the number of lost to follow-up and/or other MDR-TB treatment outcomes.

## Risk of bias in individual studies

The quality assessment appraisal was performed by two independent assessors (AA and SZ) using the standardized Joanna Briggs Institute (JBI) critical appraisal tool prepared for cross-sectional studies and cohort studies [16–18]. The tools have 'Yes', 'No', 'Unclear' or 'not applicable' responses, and scores were given 1 for 'Yes', 0 for 'No' and 'Unclear' responses. Scores for each item were summed up and transformed into percentages. The average score of the two independent assessors were taken. Only studies that scored ≥50% were considered for systematic review and meta-analysis.

## Summary measures

For the prevalence review the effect measure was prevalence of LTFU which is the percentage of participants with LTFU among MDR-TB patients (n/N*100). The Odds Ratio (OR) with its 95% confidence interval was the primary measure of effect size for the associated factors.

## Methods of analysis

The extracted data were exported to STATA/SE V.14 for the meta-analysis. The existence of heterogeneity among studies was assessed using the forest plot, the Cochrane Q statistics and the $I^2$. The forest plot provides a visual inspection of the confidence intervals of effect sizes of individual studies. The presence of non-overlapping intervals suggest heterogeneity. Significance of heterogeneity was declared using Q statistics at p-value <0.1 [19]. Whereas, $I^2$ values of 25%, 50% and 75% were considered as low, moderate, and high heterogeneity, respectively [20]. The Freeman-Tukey (double arcsine) transformation was performed to avoid the large weight gained by studies with extreme prevalence [21,22]. The confidence intervals were computed using the exact method [22]. The DerSimonian and Laird (D-L) method for the random effects model was applied in the meta-analysis of the prevalence and associated factors of lost to follow-up [23].

## Risk of bias across studies

Funnel plot was used to detect and examine publication and small study biases. The funnel plot asymmetry was statistically checked using Egger's test [24] and Begg's test [25]. Accordingly, asymmetry of the funnel plot and/or statistical significance of Egger's regression test and Begg's rank correlation test (p<0.05) were suggestive of publication or small study bias.

## Additional analyses

Subgroup analysis was performed by using the region and publication year as grouping variables and sources of variation. Meta regression was also conducted for the prevalence review using study characteristics such as region, sample size and mean age as covariates. Sensitivity analysis was carried out for the prevalence meta-analysis but not for the associated factors due to small number of included studies.

## Results

Study selection

A total of 1,198 studies were found through electronic searches. About 52 full text articles were assessed for eligibility. Twelve studies met the inclusion criteria and were included in the qualitative synthesis. But, only 11 studies were eligible for meta-analysis (Fig 1).

## Study characteristics

From the twelve selected studies for qualitative synthesis, 9 studies were retrospective cohort [26–34], 2 cross sectionals [35,36] and 1 prospective cohort study [37]. The publication year range from 2014 to 2020. Regarding the study area, 3 studies were conducted in Amhara region [26,27,30], 3 in Oromia region [28,33,34], 1 in Addis Ababa [35], 1 in Southern Nations, Nationalities and People (SNNP) [29] and the rest four were conducted in multiple regions of the country [31,32,36,37]. The study period ranged from 3 years to 10 years. These studies were conducted among 3,923 participants. The smallest and largest sample sizes were 136 [33] and 680 [35], respectively. This review was concerned only about participants whose final treatment outcome were ascertained and hence excluded those participants who were on treatment during the study periods or who were transferred out to other treatment centers. Therefore, after excluding patients still on treatment and transferred out, the effective sample size ranged from 90 [27] to 612 [37] and comprised a total of 3,510 participants. These effective

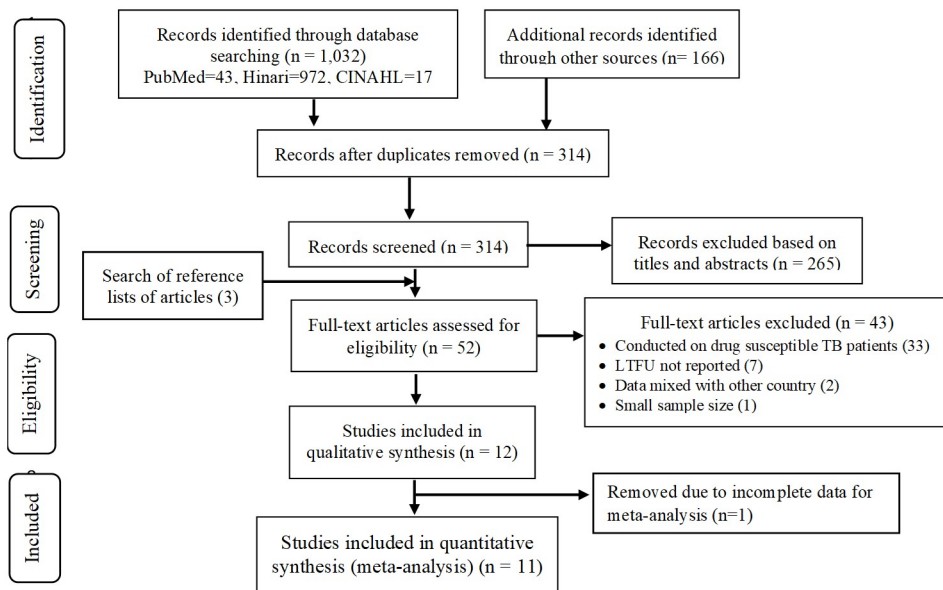

**Fig 1. PRISMA flow diagram of included studies in the systematic review and meta-analysis of lost to follow-up and associated factors among patients with MDR-TB in Ethiopia from 2014–2020.**

**Table 1. The characteristics of studies included in the systematic review and meta-analysis on lost to follow-up and associated factors among patients with MDR-TB in Ethiopia from 2014–2020.**

| Author | Pub. Year | Region | Study period | Study design | Sample size | Effective sample* | Statistical model | Mean Age | Primary outcome |
|---|---|---|---|---|---|---|---|---|---|
| Alene et al [26] | 2017 | Amhara | 2010–2015 | Retro. Cohort | 242 | 216 | Cox prop.haz | 34 | Poor outcome |
| Baye et al [27] | 2018 | Amhara | 2012–2016 | Retro. Cohort | 141 | 90 | Chi-square | 30.87 | Successful outcome |
| Fantaw et al [28] | 2018 | Oromia | 2013–2017 | Retro. Cohort | 164 | 142 | Cox prop.haz | 31.5 | Death |
| Girum et al [29] | 2017 | SNNP | 2013–2017 | Retro. Cohort | 154 | 98 | Cox prop.haz | 28 | Death |
| Kassa et al [30] | 2019 | Amhara | 2010–2017 | Retro. Cohort | 332 | 290 | Gompertz reg | 31 | LTFU |
| Mequanint et al [35] | 2014 | A/A | 2011–2013 | Cross sectional | 680 | 550 | Logistic reg. | 30 | Successful outcome |
| Meressa et al [37] | 2015 | Multiple | 2009–2014 | Prosp. Cohort | 612 | 612 | Cox prop.haz | 28.3 | Poor outcome |
| Molla et al [36] | 2017 | Multiple | 2012–2014 | Cross sectional | 178 | 176 | Descriptive | N/R | All outcomes |
| Shimbre et al [31] | 2020 | Multiple | 2009–2016 | Retro. Cohort | 462 | 404 | Cox prop.haz | 28.7 | Death/failure |
| Tola et al [32] | 2019 | Multiple | 2009–2019 | Retro. Cohort | 407 | 407 | Poisson reg | 31.8 | Successful outcome |
| Wakjira et al [33] | 2019 | Oromia | 2012–2016 | Retro. Cohort | 136 | 110 | Logistic reg. | 32.12 | Favorable outcome |
| Woldeyohannes et al [34] | 2019 | Oromia | 2012–2017 | Retro. Cohort | 415 | 415 | Cox prop.haz | 28 (median) | Unfavorable outcome |

*Sample size after the ineligible population was reduced, i.e., MDR-TB patients who were still on treatment or transferred out. Since the final treatment outcome cannot be ascertained for these population, we reduced from the calculation of the prevalence of LTFU.

N/R: Not Reported.

A/A: Addis Ababa.

sample sizes were the samples which were used as a denominator for the calculation of the prevalence of LTFU. Regarding the primary outcome of included studies, four studies determined successful (favorable) treatment outcome [27,32,33,35], three studies poor (unfavorable) treatment outcomes [26,34,37], three studies focused on death [28,29,31], one study determined LTFU [30], and one study determined all treatment outcomes [36]. In all studies, the proportion of LTFU was either separately reported or could be computed from the available data (Table 1).

## Risk of bias within studies

The JBI check list was used to assess the quality of individual studies. The checklists for prevalence, analytical cross-sectional and cohort studies were used with the respective study designs. The scoring was performed by assigning 1 for yes, 0 for no and uncertain. Accordingly, all selected studies fulfilled the 50% quality assessment score for the qualitative synthesis. However, one study was excluded for the meta-analysis due to the fact that the outcome variable (LTFU) was not clearly reported (S2 File).

## Synthesis of results

**Prevalence of lost to follow-up.** Based on the eligibility criteria 11 studies were selected for meta-analysis of the prevalence of LTFU. The pooled random effects prevalence was 8.66% (95% CI 5.01–13.14; p < 0.001). There was a high heterogeneity between studies as evidenced by a significant heterogeneity chi-squared statistic (Q = 153.50 (d.f. = 10), p <0.001) and $I^2$ = 93.49% with p<0.001. The estimate of between-study variance (Tau-squared) was 0.05 (Fig 2).

**Sub group analysis by region.** From the subgroup analysis performed by region, there was no heterogeneity among studies conducted in Amhara, Oromia and other (A/A and

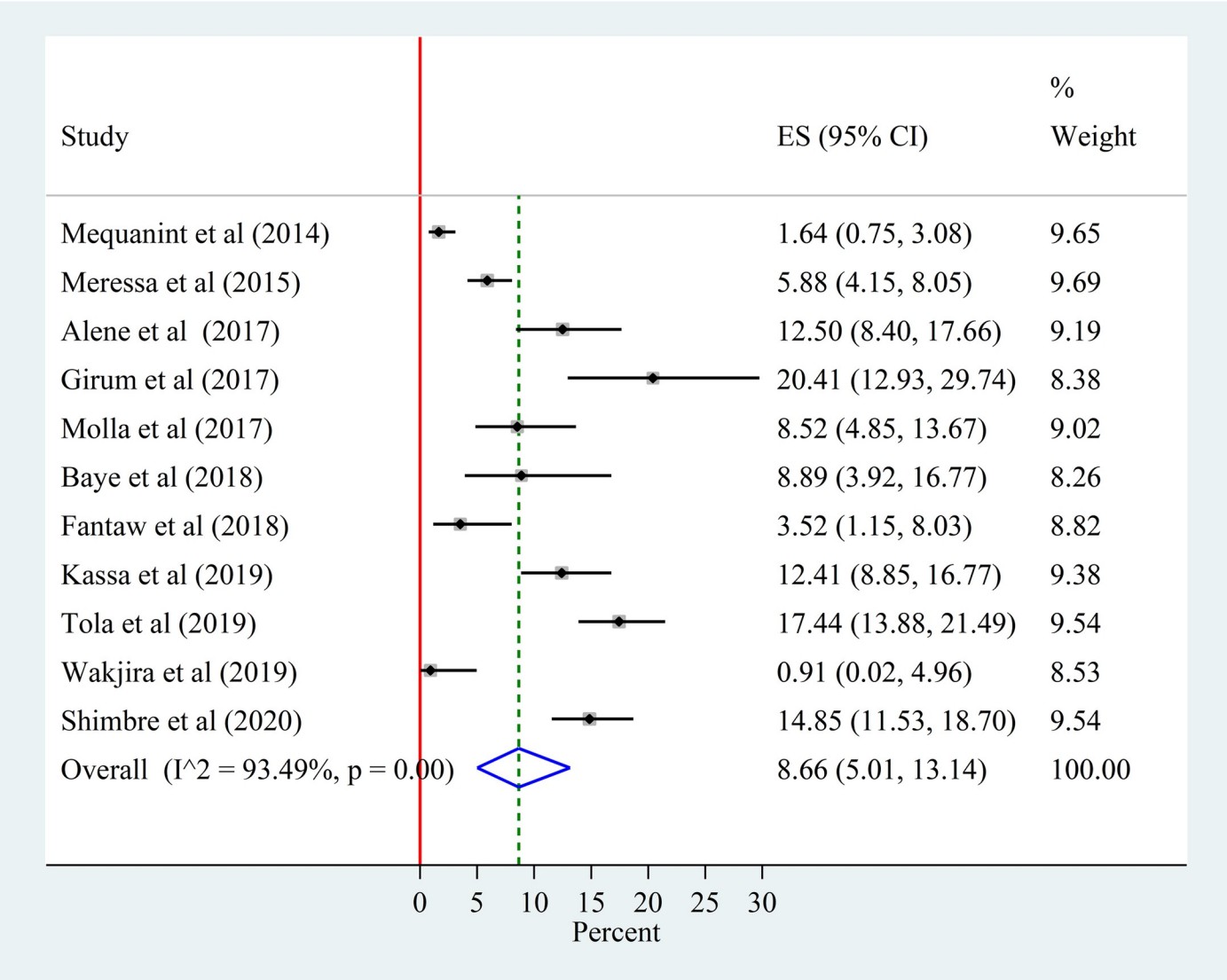

**Fig 2. Forest plot of the pooled prevalence of lost to follow-up among patients with MDR-TB in Ethiopia from 2014–2020.**

SNNP) regions. But there is still high heterogeneity among studies conducted at least in multiple regions. The prevalence of LTFU was 11.84% (95% CI, 6.33–14.58), 2.21% (95% CI, 0.63–4.53), 11.27% (95% CI, 6.00–17.90), and 3.05% (95% CI, 1.81–4.57) in Amhara, Oromia, multiple regions and others, respectively (Fig 3).

**Subgroup analysis by year of publication.** There was no difference in the prevalence of LTFU between studies published before 2017 and after 2017. The within group heterogeneity was high in both groups. The sub-groups pooled prevalence was also similar with the overall pooled prevalence. That means the overall heterogeneity was not explained by publication year (Fig 4).

**Meta regression.** We further investigated the heterogeneity using different statistical techniques to identify the source of heterogeneity. A meta-regression was performed using region, sample size and mean age as covariates and by specifying the REML method for estimating the

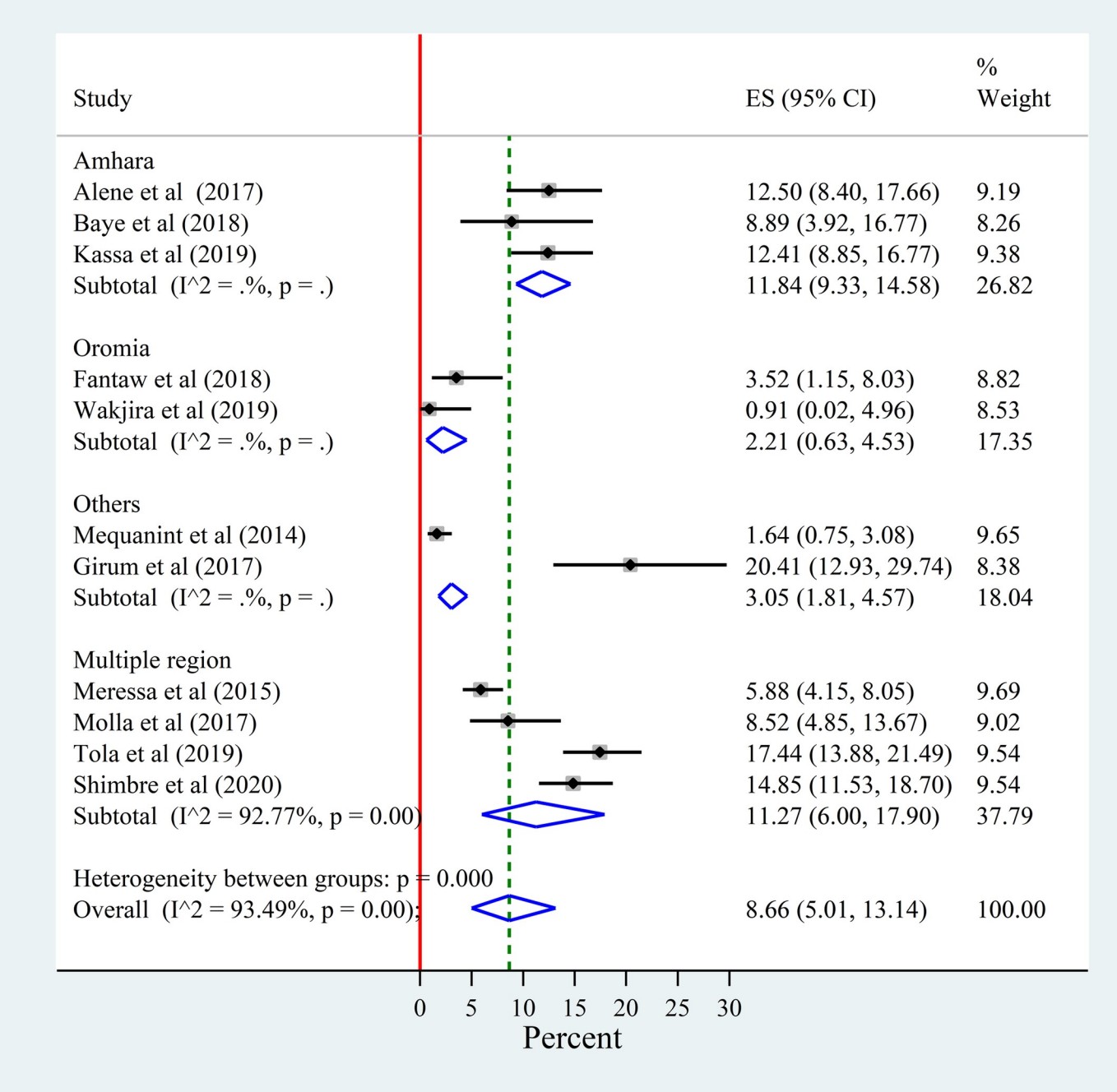

**Fig 3. Forest plot for the sub-group analysis of the prevalence of lost to follow-up among patients with MDR-TB by study region in Ethiopia from 2014–2020.**

between-study variance. None of these variables were statistically significant for explaining the heterogeneity (Table 2).

**Sensitivity analysis.** To check the influence of a single study on the effect size, a sensitivity analysis was performed using the random effects model and none of the studies had significant influence (Fig 5).

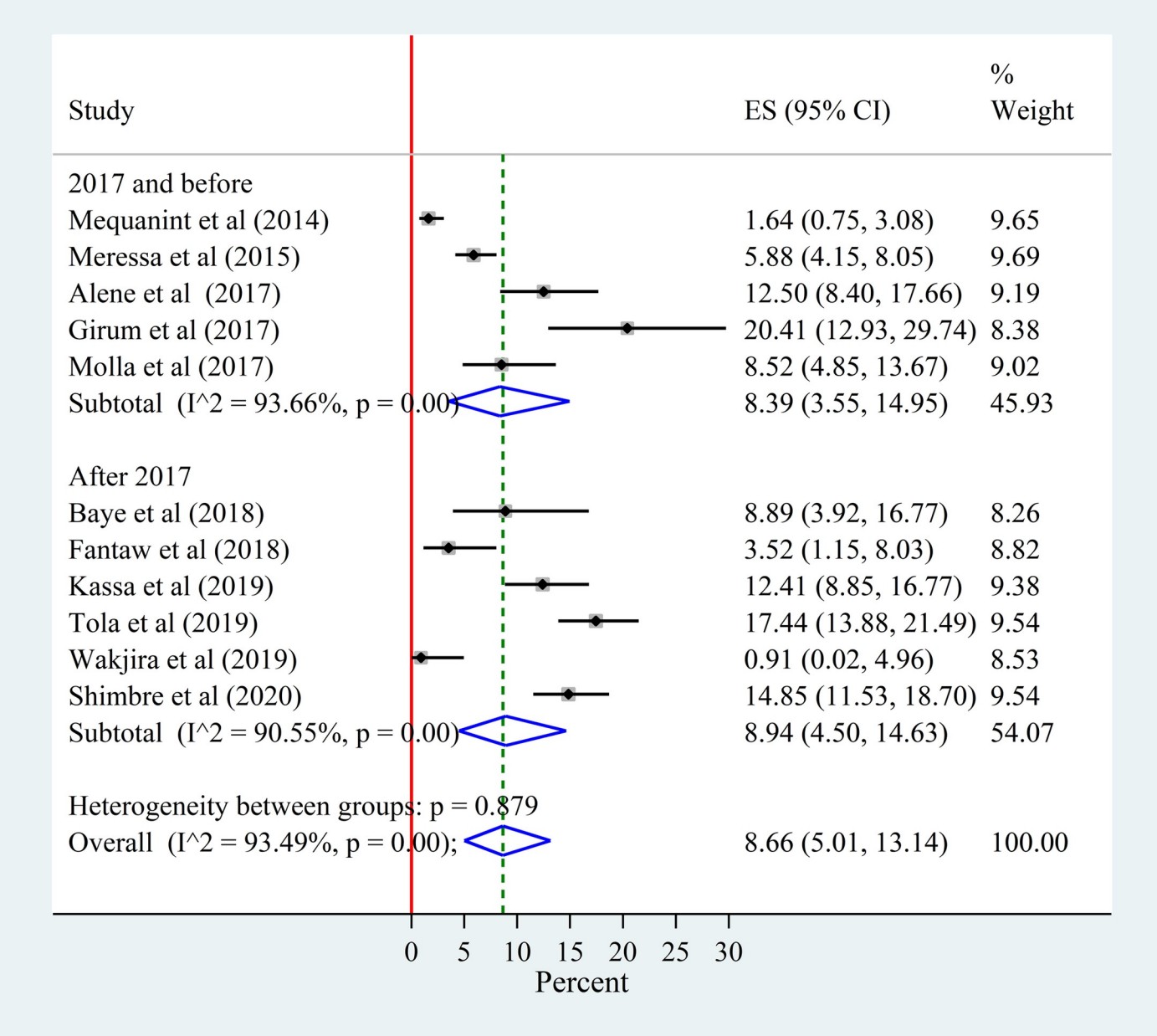

**Fig 4. Forest plot for the sub-group analysis of the prevalence of lost to follow-up among patients with MDR-TB by publication (study) year in Ethiopia from 2014–2020.**

### Risk of bias across studies

The presence of publication bias was assessed using funnel plot, Egger's and Begg's statistical tests at 5% level of significance. The funnel plot was performed by labeling the prevalence (the effect size) to the x-axis and standard error of the prevalence to the y-axis. There was no a significant publication or small study effect as evidenced by symmetrical funnel plot and insignificant Egger's test (p = 0.670) and Begg's test (p = 0.815) (Fig 6).

**Table 2. Factors related with heterogeneity of the prevalence of lost to follow-up among patients with MDR-TB in Ethiopia, 2014–2020.**

| Variables | Coefficients (95%CI) | p-value |
|---|---|---|
| Amhara | 0.0040892 (-0.1186345, 0.1268129) | 0.938 |
| Oromia | -0.0910473 (-0.2611571, 0.0790624) | 0.238 |
| Addis Ababa | -0.0984228 (-0.229194, 0.0323485) | 0.115 |
| SNNP | 0.0892953 (-0.1695685, 0.348159) | 0.431 |
| Multiple regions | Reference | |
| Sample size | -0.0000943 (-0.0003558, 0.0001673) | 0.436 |
| Mean age | 0.0024965 (-0.0282739, 0.0332669) | 0.856 |

## Factors associated with lost to follow-up

Data were obtained only from 4 studies [26,28,30,32] for the identification of the factors associated with LTFU. Other studies were not included because the information to compute the effect sizes were not available. From these studies, only one [30] was primarily conducted to determine the factors associated with lost to follow-up. The rest three [26,28,32] were included after the necessary data were obtained through e-mail communication with authors. The meta-analysis was performed to determine the effects of five variables including sex, HIV status, history of previous treatment, comorbidities and anatomical site of TB. From the five

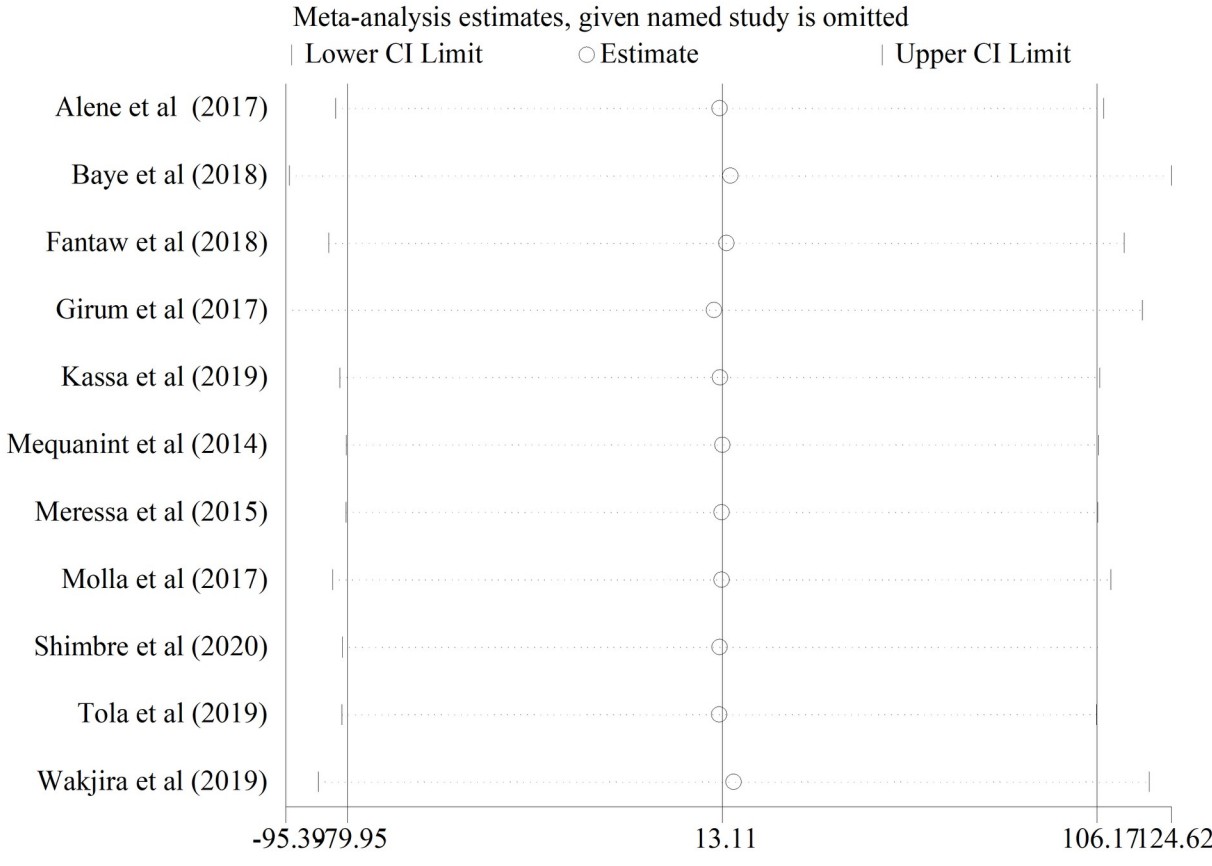

**Fig 5. Sensitivity analysis for single study influence on the prevalence of LTFU in Ethiopia from 2014–2020.**

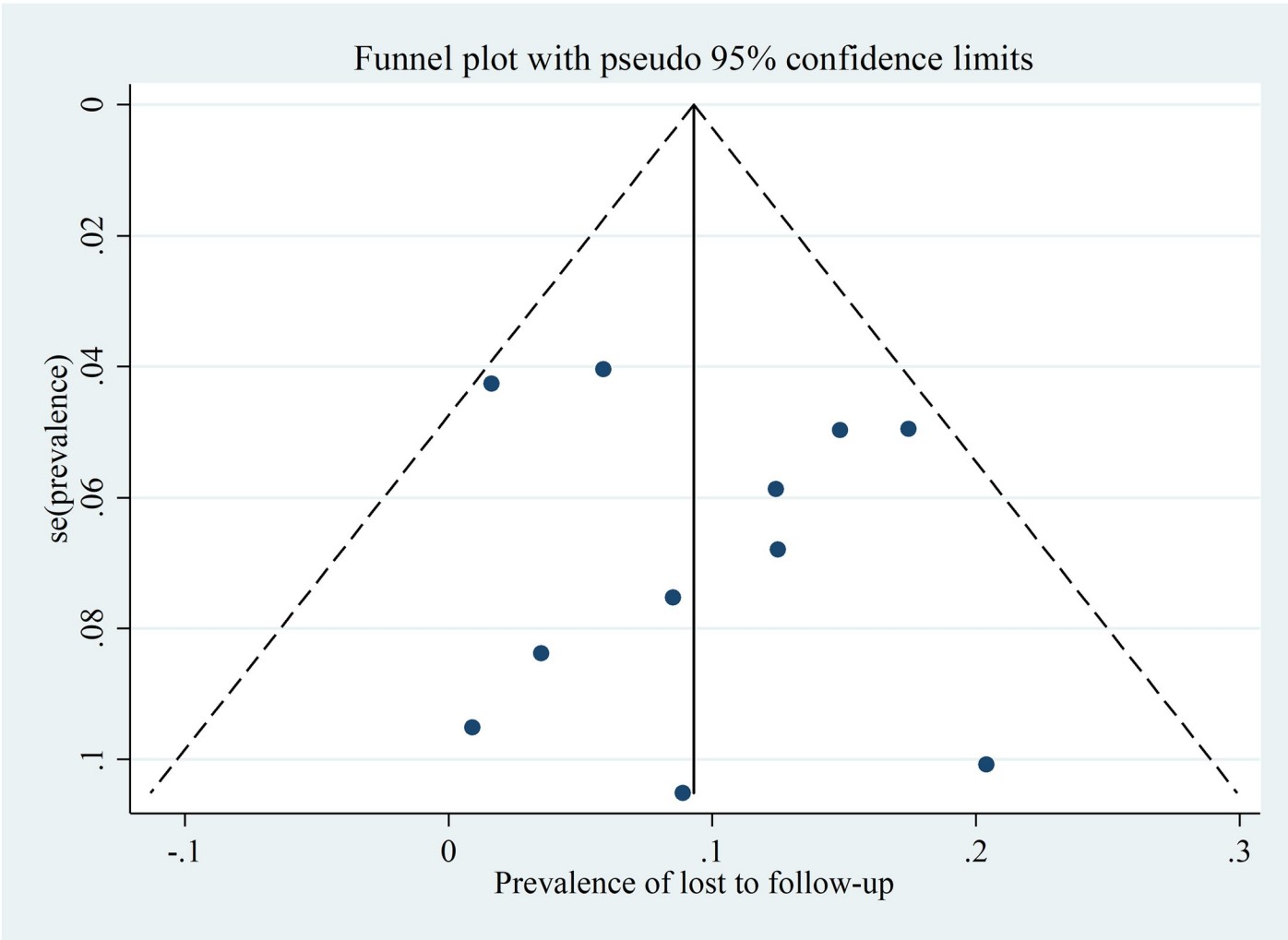

**Fig 6. A funnel plot for publication bias of lost to follow-up among patients with MDR-TB in Ethiopia from 2014–2020.**

variables, only one (anatomical site of TB) had statistically significant effect on the prevalence of LTFU. Additional analysis like subgroup analysis, meta regression and sensitivity analysis were not performed due to the small number of studies included.

**Effect of anatomical site of MDR-TB on lost to follow-up.** The odds of LTFU among patients with pulmonary MDR-TB was 50% less compared to patients with extra-pulmonary MDR-TB (OR = 0.50, 95% CI 0.27,0.96). The included studies had no significant heterogeneity ($I^2 = 0$, p = 0.912) (Fig 7).

**Risk of bias across studies used in the identification of the associated factors.** A funnel plot was plotted by labeling the log of the effect size (log OR) to the x-axis and the standard error of the log (OR) to the y-axis. There was no significant publication or small study effect as evidenced by the symmetrical funnel plot, insignificant Egger's test (p = 0.633) and Begg's test (p = 0.174) (Fig 8).

## Discussion

This systematic review and meta-analysis were aimed at estimating the pooled prevalence and associated factors of lost to follow-up among drug resistant TB patients in Ethiopia.

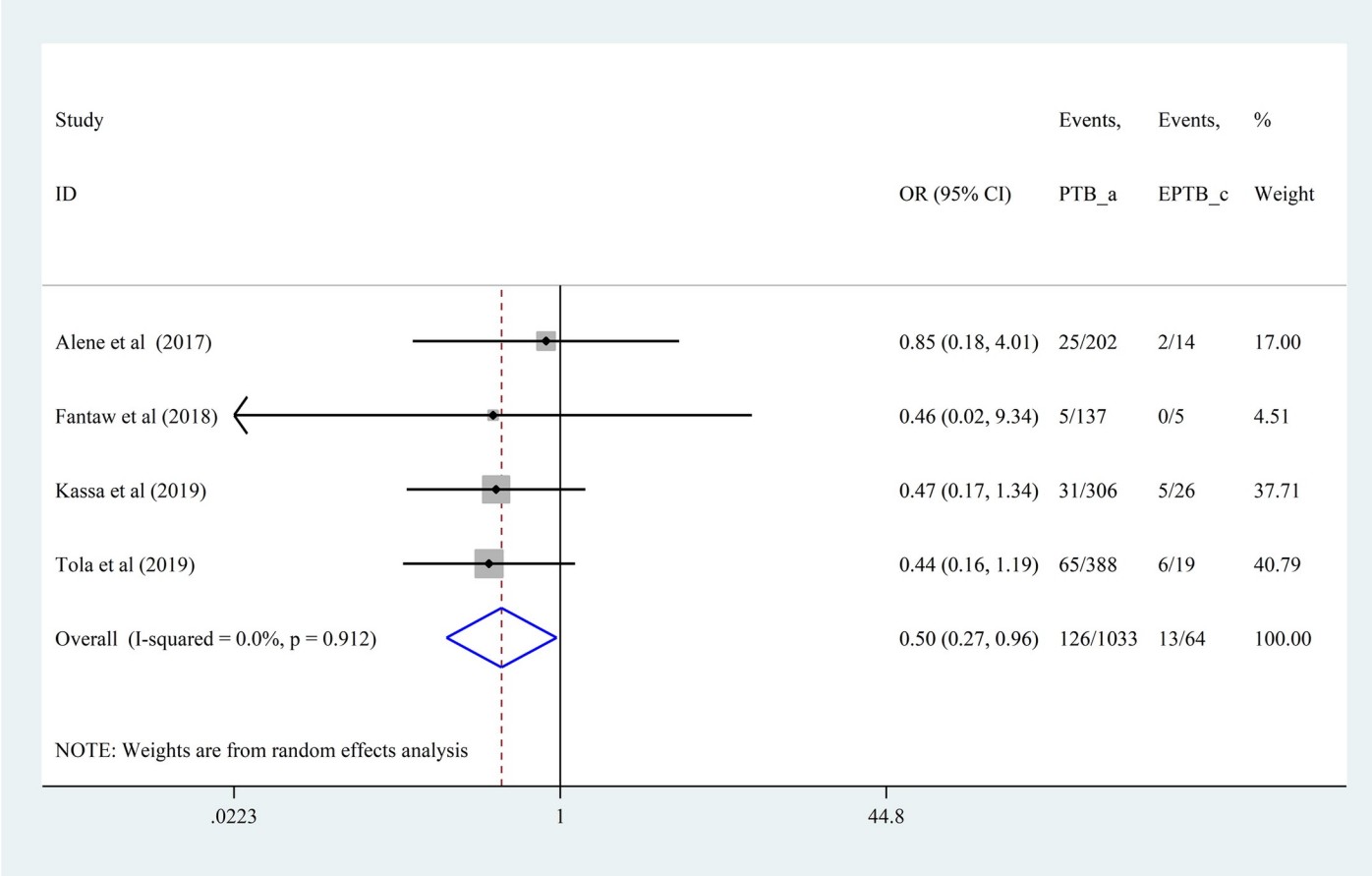

**Fig 7. Forest plot of the pooled effect of anatomical site of TB on lost to follow-up among patients with MDR-TB in Ethiopia from 2014–2020.**

The pooled estimate of the prevalence of lost to follow-up was found to be 8.66%. This estimate was considerable compared with the maximum global target of MDR/RR-TB unsuccessful treatment outcome which is aimed to be at most 10% [1]. Here it is good to remember that unsuccessful treatment outcome consists of interruption, LTFU, failure and death. Even if we did not get a separate target for LTFU, this pooled prevalence can be considered as high relative to the global target of unsuccessful treatment outcome. This prevalence of LTFU implies that a significant number of DR-TB patients was discontinued from their recommended treatment before they reached to the recommended duration. These people eventually live with the community signifying a high rate of transmission of the disease within the public. On the other hand, these defaulted patients are at a higher risk of progressing to the next level of DR TB and death [12]. Therefore, the burden of DR-TB in the country will escalate as the prevalence of lost to follow-up increases.

The sub group analysis indicated that there was a variation in the prevalence of LTFU across regions in the country. The highest prevalence of lost to follow-up was observed in Amhara region which was 11.84%. This is a signal for the Amhara region to work hardly on reducing LTFU and further investigation of why the region has higher LTFU is important.

The anatomical site of TB was significantly associated with the prevalence of LTFU. Those patients with pulmonary TB were 50% less likely to loss from treatment follow-up than patients with extra pulmonary TB. This difference may be attributed to the nature of the

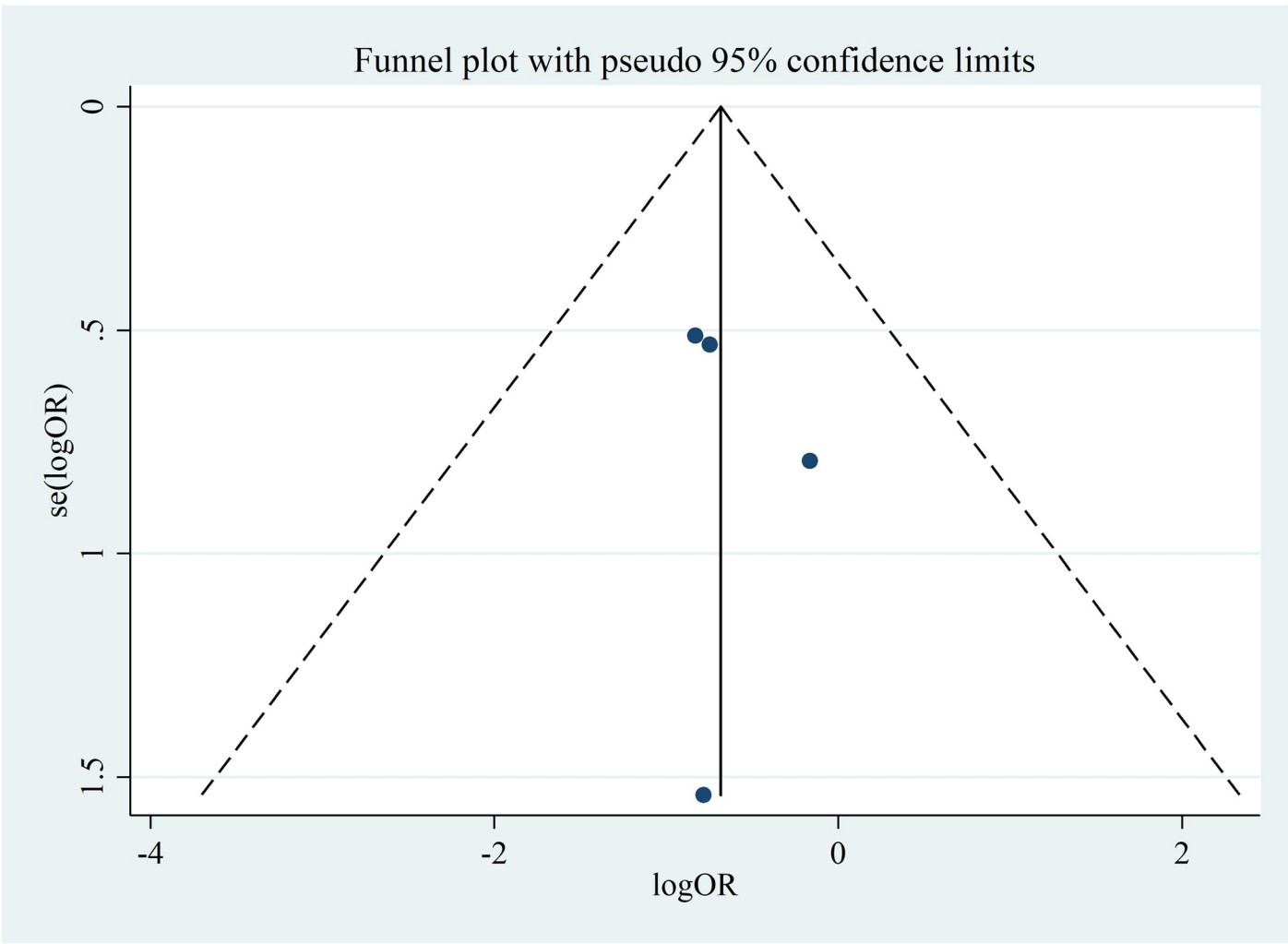

**Fig 8. A funnel plot for publication bias of the effect of anatomical site of TB on lost to follow-up among patients with MDR-TB in Ethiopia from 2014–2020.**

disease and the treatment regimens followed in these patients. Patients with PTB are more symptomatic than patients with EPTB [38]. As a patient shows signs and symptoms, that patient may develop an illness psychology which in turn initiates the patient to accept advises from the health care workers in the urge to avoid signs and symptoms. This may help PTB patients to adhere to the recommended duration of TB treatment compared to EPTB patients. On the other hand, most pulmonary TB patients are managed using the shorter treatment regimen lasting 9–12 months. Compared to the extra pulmonary TB patients by which most of them are managed with the longer treatment regimen (18–20 months), pulmonary TB patients had a lesser burden of treatment helping them to complete the whole duration [5]. This may alert health care providers to follow extra pulmonary TB patients with a better emphasis than PTB patients.

Other factors like sex, HIV status, history of previous treatment, and comorbidities were not significantly associated with LTFU. But this does not mean that these variables had no effect on LTFU rather the current studies may not be sufficient to detect the effect of these factors.

## Strengths of the study

Unlike previous reviews, this review included both published and unpublished studies. In addition, final outcomes were measured from eligible population i.e., patients who completed their treatment regimen duration.

## Limitations of the study

This review determined the prevalence of lost to follow-up in the presence of high heterogeneity among individual studies which limits the direct interpretation of pooled estimates. In addition, factors were not explored adequately due to the lack of data among individual studies. Due to this, small number of studies were included in the analysis of the effect of few factors associated with lost to follow-up and this ended up with insignificant effect sizes across the included variables. Knowing the moment when patients lost from follow-up could also be useful to evaluate the influence of the length of treatment and to identify the moment of greatest risk of LTFU in both presentations of the disease (PTB & EPTB). However, the data available from the included studies was not sufficient to fill this gap. Moreover, the issue of generalizability was in question since the studies were not from all regions of the country.

## Conclusions

The prevalence of lost to follow-up among MDR-TB patients in Ethiopia was high with observed differences across regions. The anatomical site of MDR-TB was significantly associated with lost to follow-up. Strengthening patient centered monitoring system to tackle the obstacles of treatment compliance both on the patient side and in the health care system is important to reduce LTFU. Health care professionals working in MDR-TB treatment centers should provide counseling on treatment continuation giving a special emphasis to extra pulmonary MDR-TB patients. One of the challenges we encountered in this review were lack of studies conducted exclusively on lost to follow-up. Thus, researchers are recommended to conduct primary studies focusing lost to follow-up as a primary outcome. Moreover, it is good to conduct individual patient data (IPD) meta-analysis to get pooled estimates of effect sizes for the associated factors using currently available primary studies.

## Supporting information

**S1 File. The search strategies for selected data bases.**
(DOCX)

**S2 File. The quality assessment score of included studies.**
(DOCX)

**S3 File. PRISMA 2009 checklist.**
(DOC)

**S4 File. Review protocol.**
(DOCX)

**S1 Dataset. Prevalence of LTFU.**
(DTA)

**S2 Dataset. Factors associated with LTFU.**
(DTA)

## Acknowledgments

We acknowledge authors of primary studies who provided us additional information for the meta-analysis.

## Author Contributions

**Conceptualization:** Assefa Andargie.

**Data curation:** Assefa Andargie, Segenet Zewdie.

**Formal analysis:** Assefa Andargie, Segenet Zewdie.

**Investigation:** Assefa Andargie.

**Methodology:** Assefa Andargie, Asressie Molla, Fentaw Tadese, Segenet Zewdie.

**Project administration:** Assefa Andargie.

**Resources:** Assefa Andargie.

**Software:** Assefa Andargie.

**Supervision:** Asressie Molla, Fentaw Tadese.

**Validation:** Asressie Molla, Fentaw Tadese, Segenet Zewdie.

**Visualization:** Asressie Molla, Fentaw Tadese.

**Writing – original draft:** Assefa Andargie.

**Writing – review & editing:** Assefa Andargie, Asressie Molla.

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
