## [Decision Letter · Decision Letter 0]

4 Feb 2021

PONE-D-20-24675

Lost to follow-up and associated factors among patients with multi-drug resistant tuberculosis in Ethiopia: A systematic review and meta-analysis

PLOS ONE

Dear Dr. Assefa,

Thank you for submitting your manuscript to PLOS ONE. After careful consideration, we feel that it has merit but does not fully meet PLOS ONE’s publication criteria as it currently stands. Therefore, we invite you to submit a revised version of the manuscript that addresses the points raised during the review process.

We look forward to receiving your revised manuscript.

Kind regards,

Claudia Marotta

Academic Editor

PLOS ONE

Journal Requirements:

2. During our initial in-house review, we found the following reference investigating a similar research question as your study:

Prevalence of tuberculosis treatment non-adherence in Ethiopia: a systematic review and meta-analysis

https://www.ingentaconnect.com/content/iuatld/ijtld/2019/00000023/00000006/art00015

Please discuss how the current manuscript advances on this previous work in your introduction and/or discussion section.

3. We note that your quality assessment in your supplemental material as the criteria numbered, but does not state what numbers correlate to which criteria. Please add an additional page to this file, stating what each number correlates to.

4. In your methods section, please provide the publication date range that was used to find the manuscripts included in your study.

5.Thank you for stating the following financial disclosure:

 "The funders had no role in study design, data collection and analysis, decision to publish, or preparation of the manuscript"

6. We note you have included a table to which you do not refer in the text of your manuscript. Please ensure that you refer to Table 2 in your text; if accepted, production will need this reference to link the reader to the Table.

7. Please include a copy of Table 3 which you refer to in your text on page 15.

Additional Editor Comments:

dear authors follow reviewers suggestion to improve your paper

Reviewers' comments:

Reviewer's Responses to Questions

**Comments to the Author**

1. Is the manuscript technically sound, and do the data support the conclusions?

Reviewer #1: Yes

Reviewer #2: Yes

Reviewer #3: Yes

2. Has the statistical analysis been performed appropriately and rigorously? 

Reviewer #1: Yes

Reviewer #2: Yes

Reviewer #3: Yes

3. Have the authors made all data underlying the findings in their manuscript fully available?

Reviewer #1: Yes

Reviewer #2: Yes

Reviewer #3: Yes

4. Is the manuscript presented in an intelligible fashion and written in standard English?

Reviewer #1: Yes

Reviewer #2: Yes

Reviewer #3: Yes

5. Review Comments to the Author

Reviewer #1: 8% loss to follow up is not a large proportion for patients on second line if compared to global treatment outcomes.

Two very minor things need to be revised:

1- There is a misconception about the term cohort studies and cohort analysis, so there is a copied mistake in my opinion. Cohort studies look for difference in risk due to exposure, while cohort analysis for a treatment group focuses on treatment outcomes.

2- References writing need to be revised, e.g. reference no.29 did not identify the journal

Reviewer #2: It's a very interesting and important strudy. But... there are some aspects to review.

1: page 15 line 282: may to say "Table 2".

2. Figure 5: doesnt exist...

3. In Introduction there are 13 articles of the bibliography, but in the discussion you have only used 2 articles. I think the Discussion may have your results compared or analyzed against some of the results obtaines in the studies you have analyzed.

Good work....

Reviewer #3: Although the authors have included few articles (11) at the end of the PRISMA selection, they have used techniques of verification to rule out possible publication biases and have managed well to estimate the prevalence of LTFU in Ethiopia.

The authors also concluded that patients with extrapulmonary forms have greater LTFU than patients with pulmonary forms. They also reported that there are three types of therapeutic regimens, and that patients with extrapulmonary forms use longer regimens.

The authors attribute LTFU to the clinical presentation of TB. Considering that extrapulmonary forms have a longer pre-diagnostic morbidity period, which could compensate for the willingness to complete the treatment, I believe that a discussion on the reasons for LTFU should be further explored, especially the burden of pills and the length of treatment.

Knowing the moment when patients LTFU of the treatment can also be useful to evaluate the influence of the length of treatment and to identify the moment of greatest risk of LTFU in both presentations of the disease.

Maybe the data available is not enough to fill these recommendation. Then I suggest the authors to include as a limitation of the study.

6. PLOS authors have the option to publish the peer review history of their article (what does this mean?). If published, this will include your full peer review and any attached files.

Reviewer #1: **Yes: **Layth Al-Salihi

Reviewer #2: No

Reviewer #3: No

---

## [Author Response · Author response to Decision Letter 0]

10 Feb 2021

Response to points raised by the academic editor

1. We have checked each section and all are inline with the PLOS ONE’s style requirement. 

2. We have discussed how the current manuscript advances on the stated previous work in the introduction section of the manuscript (page 5, line 92-100)

3. We included an additional text on the previous document to correlate each number to the listed assessment criteria (S2_file)

4. The publication date range used to find articles included in the methods section (page 7, line 129).

5. Financial disclosure stated as ‘The authors received no specific funding for this work’ (page 21, line 394)

6. The cited table i.e., ‘Table 3’ in the text was replaced by ‘Table 2’ (page 15, line 284).

7. The cited table was wrong (editorial problem) and corrected as stated in number 6 above.

Response to reviewers’ comments

Reviewer #1

• The 8% LTFU was stated as high by comparing it with the aspired global target of 90% treatment success. It is corrected as considerable (page 18, line 332).

1. The study design labeled as ‘cohort’ was directly taken from the articles. As the reviewer raised, there is no any true cohort study that compared exposed and unexposed groups with respect to the outcome. But there are cohort analyzed studies focusing only a single group to determine the outcome. Therefore, what we could do as systematic review is to write the design that the primary researcher wrote.

2. The references list was corrected to fit with the journal requirement (page 24-25, line 475-489). 

Reviewer #2

1. The cited table was changed from Table 3 to Table 2 (page 15, line 284)

2. Figure 5 was available. We do not know whether the copy sent to this reviewer had a problem. Still, it is attached as a separate file labeled as ‘Fig5’.

3. Regarding few numbers of articles cited in the discussion while there were 13 references in the introduction, 

Most of the citations in the introduction were global reports and documents other than research articles. There was a shortage of similar reviews on LTFU which could be used in the discussion section. That is why we compared our finding the global target. Therefore, it is due to the lack of similar findings in line with our review that forced us to cite only 2 references in the discussion section. We have added 2 more references in the discussion to support our explanation about the finding (page 18, line 340 and page 19, line 351)

Reviewer #3

We have added references that support our explanation (page 18, line 340 and page 19, line 351)

In the limitation section, we added the importance of knowing the moment of LTFU and our inability to include this information in our analysis (page 20, line 373-376).

---

## [Decision Letter · Decision Letter 1]

4 Mar 2021

Lost to follow-up and associated factors among patients with drug resistant tuberculosis in Ethiopia: A systematic review and meta-analysis

PONE-D-20-24675R1

Dear Dr. Assefa,

We’re pleased to inform you that your manuscript has been judged scientifically suitable for publication and will be formally accepted for publication once it meets all outstanding technical requirements.

Kind regards,

Claudia Marotta

Academic Editor

PLOS ONE

Additional Editor Comments (optional):

dear authors congratulations

Reviewers' comments:

Reviewer's Responses to Questions

**Comments to the Author**

1. If the authors have adequately addressed your comments raised in a previous round of review and you feel that this manuscript is now acceptable for publication, you may indicate that here to bypass the “Comments to the Author” section, enter your conflict of interest statement in the “Confidential to Editor” section, and submit your "Accept" recommendation.

Reviewer #1: All comments have been addressed

Reviewer #2: (No Response)

2. Is the manuscript technically sound, and do the data support the conclusions?

Reviewer #1: Yes

Reviewer #2: Yes

3. Has the statistical analysis been performed appropriately and rigorously? 

Reviewer #1: Yes

Reviewer #2: Yes

4. Have the authors made all data underlying the findings in their manuscript fully available?

Reviewer #1: Yes

Reviewer #2: Yes

5. Is the manuscript presented in an intelligible fashion and written in standard English?

Reviewer #1: Yes

Reviewer #2: Yes

6. Review Comments to the Author

Reviewer #1: The authors perfectly defended themselves and gave appropriate revisions. The study is sound for publication

Reviewer #2: Wee think this study contribute to the knowledge of the TB situation in Ethiopia and can help to improve the TB programme.

The document have two versions of the study. I think the last with modifications is the corrected for publication

7. PLOS authors have the option to publish the peer review history of their article (what does this mean?). If published, this will include your full peer review and any attached files.

Reviewer #1: **Yes: **Layth Al-Salihi

Reviewer #2: No

---

## [Editor Report · Acceptance letter]

8 Mar 2021

PONE-D-20-24675R1 

Lost to follow-up and associated factors among patients with drug resistant tuberculosis in Ethiopia: A systematic review and meta-analysis 

Dear Dr. Andargie:

I'm pleased to inform you that your manuscript has been deemed suitable for publication in PLOS ONE. Congratulations! Your manuscript is now with our production department. 

Kind regards, 

on behalf of

Dr. Claudia Marotta 

Academic Editor

PLOS ONE